# Predictive Factors of Immunotherapy in Gastric Cancer: A 2024 Update

**DOI:** 10.3390/diagnostics14121247

**Published:** 2024-06-13

**Authors:** Vasile Bintintan, Claudia Burz, Irena Pintea, Adriana Muntean, Diana Deleanu, Iulia Lupan, Gabriel Samasca

**Affiliations:** 1Department of Surgery 1, Iuliu Hatieganu University of Medicine and Pharmacy, 400006 Cluj-Napoca, Romania; vasile.bintintan@umfcluj.ro; 2Institute of Oncology, “Prof. Ion Chiricuta”, 400015 Cluj-Napoca, Romania; cristina.burz@umfcluj.ro; 3Department of Immunology, Iuliu Hatieganu University of Medicine and Pharmacy, 400006 Cluj-Napoca, Romania; nedelea@umfcluj.ro (I.P.); adriana.muntean@umfcluj.ro (A.M.); deleanu@umfcluj.ro (D.D.); 4Department of Molecular Biology, Babes-Bolyai University, 400084 Cluj-Napoca, Romania; iulia.lupan@ubbcluj.ro

**Keywords:** predictive factors, immunotherapy, gastric cancer, update

## Abstract

Many studies on gastric cancer treatment have identified predictors of immunotherapy benefits. This article provides an update on the major developments in research related to predictive factors of immunotherapy for gastric cancer. We used the search term “predictive factors, immunotherapy, gastric cancer” to find the most current publications in the PubMed database related to predictive factors of immunotherapy in gastric cancer. Programmed cell death, genetic, and immunological factors are the main study topics of immunotherapy’s predictive factors in gastric cancer. Other preventive factors for immunotherapy in gastric cancer were also found, including clinical factors, tumor microenvironment factors, imaging factors, and extracellular factors. Since there is currently no effective treatment for gastric cancer, we strongly propose that these studies be prioritized.

## 1. Introduction

As a major therapeutic approach for gastric cancer (GC), immunotherapy has developed over time. However, since there are still many unanswered questions regarding GC immunotherapy, extreme caution is necessary. The underlying mechanisms of the immune response are incredibly intricate and poorly understood. In GC, there are legitimate expectations for the development of secure and more efficient immunotherapies [1,2]. Patients with GC may survive longer because of immunotherapy. Not every patient, however, survives longer. To discover patients who can benefit from immunotherapy and extend their lives, we need straightforward, affordable, and reliable markers [3]. An update on the key advancements in GC-therapy research is given in this article to find predictive factors of immunotherapy for gastric cancer (Table 1, Table 2 and Table 3). We used the search terms “predictive factors, immunotherapy, gastric cancer” to find the most recent important articles in the PubMed database. As previously mentioned, the only publications that fulfilled the inclusion criteria were from 2023 to 2024. Articles on case reports were not included in the research because they did not pertain to our study.

## 2. Programmed Cell Death Predictive Factors

Disulfide stress induces a novel type of cell death named disulfidptosis, which is characterized by the intracellular build-up of disulfides, which causes the collapse of F-actin and cytoskeleton proteins. Clinical data and risk indicators were integrated to create a nomogram, which showed excellent predictive accuracy for one-, three-, and five-year survival rates. The prognosis of GC patients may be reliably predicted by the disulfidptosis-related gene prognostic risk model [4].

A novel type of cell death called pyroptosis, apoptosis, and necroptosis (PAN)optosis has been proposed. It is distinguished by pyroptosis, apoptosis, and necroptosis, although none of these can fully account for the death of the cell. The low expression of immunological checkpoints, elevated expression of transforming growth factor beta (TGF-β), dense infiltration of fibroblasts which are cancer-associated, and M2 macrophages (alternatively activated macrophages) were strongly correlated with a high PANscore. A guide for patient classification is provided by the characterization of PANoptosis patterns, not only in GC but also in other malignancies [5].

Copper-induced cell death, or cuproptosis, is a term used to characterize the negative effects of high copper concentrations on biological functions. GC prognosis may be directly impacted by copper-induced cell death. Long noncoding RNAs (lncRNAs) with steady structures can affect cancer prognosis. Based on copper cell death-related lncRNAs (CRLs), AC129926.1, AC023511.1, AP002954.1, LINC01537, and TMEM7 were combined to create a predictive signature for GC patients. This model may be used to predict immune infiltration and the efficacy of immunotherapy [6]. By controlling autophagy, lncRNAs may play a role in the onset, progression, and treatment resistance of GC. The autophagy-related lncRNA (ARL) signature (ARLSig) is a signature of eleven autophagy-related lncRNAs (ARLs) in GC that has strong prognostic value for the survival of GC patients and may offer new targets for customized immunotherapy [7].

Programmed cell death (PCD)-related signature (PRS) was created using TCGA, GSE15459, GSE26253, GSE62254, and GSE84437 datasets through an integrative process that used ten machine learning techniques. A reduced response to immunotherapy was suggested by a higher immune escape score, exclusion tumor mutational burden (TMB) score, lower tumor immune dysfunction, and lower immunophenoscores for programmed death 1 (PD-1) and cytotoxic T-lymphocyte-associated protein 4 (CTLA4) in patients with a high PRS, lower response rate, and poor prognosis [8]. The expression of programmed death ligand-1 (PD-L1) was correlated with a greater percentage of SWItch/Sucrose Nonfermentable (SWI/SNF), lysine demethylase (KDM), phosphatidylinositol 3-kinase (PI3K), and DNA (cytosine-5)-methyltransferase (DNMT) pathway alterations (all with *p* < 0.01). Compared with wild-type GC patients, patients with mutations in the PI3K pathway had a better response rate (*p* = 0.002) and lasting clinical benefit rate with immunotherapy (*p* = 0.023, *p* = 0.038). These patients also had longer overall survival (OS) (*p* = 0,2, *p* = 0.037) and progression-free survival (PFS) (*p* = 0.084, *p* = 0.0076). Consequently, modifications to the PI3K pathway linked to PD-L1 positivity are linked to increased immunotherapy efficacy [9]. Serum cytokine levels are affected when GC patients receive chemotherapy in combination with PD-1 inhibitors. Potential valid serum predictive markers for identifying individuals who might profit from PD-1 inhibitors paired with chemotherapy include interleukin (IL)-6, IL-2, IL-17A, and the neutrophil/lymphocyte ratio (NLR) [10]. The NLR, the platelet/lymphocyte ratio (PLR), the pan-inflammatory value (PIV), and the systemic immune-inflammatory index (SII) are indicators of immunotherapy success. In advanced GC (AGC), an NLR of less than 3.65 indicates a poor prognosis after immunotherapy [3]. In patients with GC, there was a noteworthy positive association (*p* < 0.05) between low pathological differentiation and the degree of cluster of differentiation (CD) 68 expression in the tumor stroma but not in the tumor nest (TN). In the TN (*p* < 0.05), PD-L1+CD68+ macrophages were significantly associated with reduced tumor mass (<5 cm), early TNM stage (stages I+II), excellent OS, and pathological differentiation. They may also be a viable prognostic indicator in primary GC patients [11]. The number of neutrophils, macrophages, CD8+ T cells, and dendritic cells was positively associated with the degree of integrin subunit beta 1 (ITGB1) expression. Moreover, there was a correlation between PD-L1 and ITGB1 expression. For GC patients receiving anti-PD-1 medication, ITGB1 may be an effective prognostic biomarker and a useful predictor of outcome [12].

In advanced gastric adenocarcinoma with a reduced tumor mass, absence or a small number of lymph node metastases, and a large combined positive score, neoadjuvant chemotherapy plus PD-1 antibodies may be the best course of action. In patients with AGC, a nomogram model may be able to predict pathological complete response (pCR), demonstrating acceptable predictive performance and thus easing the use of individualized treatment plans [13]. A decrease in cytotoxic T-cell (CD8+ T) and CD8+ T-cell memory (CD8+ Tm) PD-1 expression, as well as the PD-1+CD8+ T/PD-1+CD4+ T-cell ratio, was found to be an independent risk factor for PFS in AGC patients receiving immunotherapy plus chemotherapy according to univariate and multivariate Cox regression analyses. In patients with AGC undergoing immunotherapy, the ratio of serum memory PD-1+CD8+ T cells to PD-1+CD8+ T/PD-1+CD4+ T cells showed a strong predictive value for response and extended survival outcomes. Memory PD-1+CD8+ T cells and the ratio of PD-1+CD8+ T cells to PD-1+CD4+ T cells may be useful for identifying individuals who would benefit from immunotherapy among AGC patients [14]. The baseline CD4+/CD8+ ratio may be a potential predictive indicator for patients under PD-1 inhibitor-based combination therapy for advanced gastric and esophageal cancer, and it can independently predict dermatological damage. A nomogram that effectively predicts OS can include the following variables: the use of antibiotics, PD-L1 expression, the CD4+/CD8+ ratio, and the Eastern Cooperative Oncology Group performance status (ECOG PS) [15]. An improved response to immunotherapy with anti-PD-1 was linked to increased expression of the major facilitator superfamily domain containing 2A (MFSD2A) in the tumor tissues of AGC patients. MFSD2A may function as a prognostic biomarker for the response to anti-PD-1 immunotherapy in AGC patients. A potential therapeutic target for improving the efficiency of anti-PD-1 immunotherapy appears to be MFSD2A, which reprograms the tumor microenvironment (TME) to support T-cell activation [16]. Lower CD4+ T, CD4+ (naive) Tn, CD4+ Tm, CD4+ (central memory) Tcm, and CD4+ (effector memory) Tem cell frequencies expressing PD-1 were found to be independent risk factors for PFS and OS in AGC patients treated with combined immunotherapy and chemotherapy according to univariate and multivariate Cox regression analyses. In AGC patients, the peripheral CD4+ T-cell subset has shown significant predictive value for therapeutic response and longer survival [17]. In GC tumor tissues, histone acetylation leads to the overexpression of HSPA4. Via the HSPA4/ALKBH5/CD58 axis, HSPA4 overexpression decreases CD58 expression in GC cells. This is followed by PD1/PDL1 activation, a reduction in CD8+ T-cell cytotoxicity, and immunological evasion of GC cells. Patients with GC who undergo surgery only have a worse OS rate when HSPA4 is upregulated. Patients with GC receiving combination immunotherapy are predicted to respond better when HSPA4 is upregulated [18]. When AGC patients are receiving PD-1 inhibitor treatment, the quantity of Treg cells is an independent factor that influences immune-related adverse events (irAEs). PFS may be impacted by irAEs, and patients may ultimately have longer PFS. The combination of Tregs, Ki-67, and age (65 years or older) can more accurately predict the likelihood of unfavorable responses [19]. The OS of patients is linked to the incidence of irAEs. The incidence of irAEs can increase a patient’s OS. The incidence of irAEs could serve as a stand-in marker for ICIs [20].

Subsets of peripheral blood lymphocytes varied in percentage in patients who responded to PD-1 inhibitors. Therefore, utilizing biomarkers based on the numbers of various lymphocyte subpopulations, it is possible to predict the clinical prognosis and efficacy of immunotherapy in patients with AGC [21]. There is debate regarding the role of cellular retinoic acid-binding protein 2 (CRABP2) in the emergence and progression of human cancers. A negative correlation between CRABP2 and the immune checkpoint markers PD-1, PD-L1, and CTLA-4 was observed [22].

## 3. Genetics Factors

Age and the risk characteristics of Golgi apparatus-related genes (GARGs) were found to be separate risk factors for GC. A nomogram that included these variables performed better in the prediction of prognosis in patients with GC [23]. In the cell cycle, transcription factors known as early 2 factors (E2Fs) are essential. According to an analysis of the immunotherapy cohort database, patients who scored higher had superior immune responses and longer survival times. For GC patients, the E2F score is a useful predictor of survival and responsiveness to treatment [24].

Two risk categories were identified from the GC patient population; the high-risk group showed elevated expression of immunological checkpoints, a greater TME, and a poor prognosis. According to the functional enrichment data, genes involved in the DNA repair pathway appeared to be enriched in the low-risk group. According to research on tumor immune dysfunction and exclusion (TIDE), GC patients in the low-risk group are likely to benefit more from immunotherapy [25]. Gene expression analysis revealed two unique GC clusters associated with positive T-cell regulation. It has been demonstrated that suppressing Dynein Axonemal Assembly Factor 3 (DNAAF3) reduces GCs’ capacity for migration and proliferation as well as their ability to activate T lymphocytes inside the TME [26].

Thirteen immune activities and sixteen immune cell types were quantified in terms of relative abundance using single-sample gene set enrichment analysis (ssGSEA). The most important gene linked to senescence that influences the development of GC is serine/threonine kinase 40 (STK40) [27]. The ssGSEA collagen score (CS) was used to assess the expression level of genes associated with collagen. It was confirmed that CS was a separate prognostic factor and that it might represent the therapeutic impact of immunotherapy [28].

By producing macrophage migration inhibitory factor (MIF), immunosuppressive microfibril-associated protein 2 (MFAP2+) cancer-associated fibroblasts (CAFs) can interact with T cells, B cells, and macrophages. This further reveals that MFAP2+ CAFs may enhance treatment resistance by controlling the malfunction of T cells and the polarization of M2 macrophages. MFAP2+ CAFs create an immune-evading TME with immune effector cells that are debilitated, which is why patients with GC are predicted to have worse clinical outcomes and a worse response to adjuvant therapy and immunotherapy [29]. MSI-XGNN, a novel computational framework for predicting MSI status using bulk RNA sequencing and DNA methylation data, identified six microsatellite instability (MSI) indicators. These indicators included two genes (MSH4 and RPL22L1) and four methylation probes (EPM2AIP1|MLH1:cg27331401, LNP1:cg05428436, and TSC22D2:cg15048832) that collectively made up the ideal feature subset. The immunotherapy characteristics of the TME, such as the tumor mutation burden, neoantigen level, and immunological checkpoint molecules, for instance PD-1 and cytotoxic T-lymphocyte antigen-4, were strongly correlated with all six markers [30].

A novel method for predicting the treatment response and stratification of patients with GC is offered by a lncRNA signature model associated with DNA methylation regulators. Analyzing the patterns of lncRNA expression in specific cancers will help us better understand how the TME gets inside the tumor and will help develop immunotherapy strategies that work better [31]. N6-methyladenosine (m6A) regulator-mediated genes in the glycolytic pathway predict the prognosis and immunotherapy response of GC patients. After reorganizing the patients, taking into account this risk model, the study successfully distinguished between the immune cell infiltration microenvironment and the immunotherapeutic response [32]. Six genes, namely, GNAS complex locus (GNAS), C-X-C motif chemokine receptor 4 (CXCR4), protein phosphatase 1 regulatory inhibitor subunit 1B (PPP1R1B), adenylate cyclase 6 (ADCY6), 5′-nucleotidase Ecto (NT5E), and nitric oxide synthase 3 (NOS3), which make up the adenosine signature, were used to stratify patients according to their risk for GC prognosis. For GC, the adenosine pathway-based signature is a potentially useful risk stratification tool that might inform personalized prognostication and immunotherapy [33].

The TIDE score and the immunophenoscore (IPS) indicate that decreased immunotherapy benefit is associated with elevated expression of glycosyltransferase (CHSY3). Experiments conducted in vitro and in vivo to enhance the invasiveness, migration, and proliferation of GC further revealed the expression of CHSY3 [34]. To predict individual results, a nomogram was created using the ferroptosis- and immunity-related gene (FIRG) signature and clinical information. FIRGs have significant therapeutic and predictive potential for GC and have a role in the onset and progression of the disease [35]. Important pathways, such as the TGF-beta, TP53, and NRF2 pathways, dominated the high-risk group, whereas the LRTK-RAS and WNT pathways represented the low-risk group. Patients with GC were effectively divided into high-risk and low-risk cohorts based on a signature of differentially expressed metabolic and immune-correlated genes (DE-MIGs), with the latter group exhibiting noticeably improved results [36].

Stomach adenocarcinoma (STAD), also known as AGC, is a prevalent type of GC with significant rates of morbidity and mortality. Anoikis-related lncRNA genes were used to construct a prognostic risk assessment model for STAD that has been shown to have high predictive accuracy and may be used as a guide for the prognostic evaluation and clinical treatment of patients with STAD [37]. A predictive model was developed, and five methylation-related genes, namely, Chromatin Assembly Factor 1 Subunit A (CHAF1A), Copine 8 (CPNE8), Pleckstrin Homology Like Domain Family A Member 3 (PHLDA3), Secreted Protein Acidic and Cysteine Rich (SPARC), and ETS Homologous Factor (EHF), were found to potentially be significant for patient prognosis. Methylation regulators, which have favorable effects for first-line clinical treatment, were used to categorize patients with STAD. The prognostic model may be able to predict a patient’s prognosis and aid in the advancement of precision medicine [38]. Hedgehog signaling, a highly conserved system that controls cell division and growth, is crucial in the development of STAD and other cancers. In STAD, the Hedgehog signaling pathway is excessively active. Hub genes in the prognostic profile of STAD, namely, growth arrest specific 1 (GAS1), GLI family zinc finger 1 (GLI1), and SCEBU2, were identified as strong risk factors for poor survival [39]. The abilities of HGC27 (a human cell line derived from the metastatic lymph node of GC) and NCI-N87 (a cell line derived from a male GC patient in 1976) cells to proliferate and migrate was severely hindered by the downregulation of secreted phosphoprotein (SPP1), which was observed following small interfering RNA (siRNA) transfection. SPP1 has been identified as a stand-alone predictor of STAD prognosis and may control the disease course by modifying the immunological milieu [40]. A new gene signature associated with T cells was created using CD5, ABCA8, SERPINE2, ESM1, SERPINA5, and NMU. With SERPINE2 boosting GC cell proliferation, the high-risk group exhibited decreased overall survival (OS), decreased immunological efficacy, and increased treatment resistance. These T-cell-related genes have been suggested as prospective targets for immunotherapy in patients with STAD and can aid in prognosis estimation [41].

Anti-PD-1/L1 monotherapy was less effective in treating Epstein–Barr virus-associated GC (EBVaGC) patients with high levels of CTLA-4. Patients with EBVaGC who received both anti-CTLA-4 and anti-PD-1/L1 checkpoint blockade profited more from combination therapy than from anti-PD-1/L1 monotherapy (*p* = 0.074). The mutation frequencies of TMB and the SWI/SNF Related, Matrix Associated, Actin Dependent Regulator of Chromatin subfamily A member 4 (SMARCA4) differed significantly between the immune checkpoint blockade (ICB)-responsive and nonresponsive groups. ICB treatment for EBVaGC may be better guided by the identification of SMARCA4, TMB, and CTLA-4 mutations as possible predictive markers of ICB success [42].

## 4. Immunological Factors

Additionally, a pan-cancer analysis disclosed a substantial association between the risk score and immunological and carcinogenic pathways. Clinical strategy design may be aided by the 4-basement membrane-related gene (BMRG) signature, which has proven accurate and reliable in the prediction of prognosis in GC patients [43].

When the percentages of M2 macrophages, myeloid-derived suppressor cells (MDSCs), and regulatory T cells (Tregs) diminished, the infiltration of cytotoxic CD8+ T cells and helper CD4+ T cells into tumors increased upon the knockdown of C-type lectin domain containing 11A (CLEC11A). In GC, CLEC11A may be a predictive and immunological biomarker. An immune signature derived from CLEC11A may offer clinicians a novel way to predict patient outcomes and create individualized treatment regimens [44]. The indicators of CD8+ T cells, neutrophils, macrophages, and dendritic cells were favorably connected with inhibitor subunit beta A (INHBA) expression, which suggests that INHBA could be a useful biomarker in the prediction of prognosis in GC patients. The INHBA is a potentially useful indicator of immunotherapy response; higher INHBA levels correspond to increased sensitivity [45].

A complement-related gene (CRG) signature based on six genes, namely, SQUAMOSA-promoter binding protein-like (SPLG), complement C9 (C9), interalpha-trypsin inhibitor heavy chain 1 (ITIH1), zinc finger protein, FOG family member 2 (ZFPM2), CD36 molecule (CD36), and serpin family E member 1 (SERPINE1), was established. In the future, in clinical practice, the unique CRG signature could be a reliable and effective tool for prognostic prediction and therapeutic guidance [46].

Multivariate analysis revealed that the natural killer (NK) cell-associated signature (NKCAS) was an independent predictive factor. Age, M stage, tumor grade, and other clinicopathological variables were combined with the NKCAS to create a nomogram that predicted patient survival. Furthermore, there was a greater infiltration of immune cells, particularly CD8+ T cells and NK cells, in the low-risk group (LRG, a 12-gene NKCAS for the Cancer Genome Atlas (TCGA) cohort, which classified GC patients into a low-risk group). The risk score and inflammatory activity were negatively correlated. Analysis of the independent immunotherapy cohort revealed that the LRG group had a much better prognosis and immunotherapy response than did the high-risk group (HRG), defined by a 12-gene NKCAS for the Cancer Genome Atlas. This study’s recognition of NK cell marker genes points to possible treatment targets. Furthermore, the nomograms and prognostic signatures that have been created may help with the clinical management of GC [47].

The expression of the C-terminal A protective factor in GC, src kinase (CSK), is correlated with immune checkpoint molecules and the degree of immune cell infiltration. It was discovered that CSK was a separate prognostic factor for GC and may predict immunotherapy and molecular targeting in addition to offering suggestions for a treatment plan [48].

Patients with GC may benefit from the use of high activin A receptor type-1 (ACVR1) expression as an independent prognostic indicator for survival. Given the strong correlation between immune infiltration and ACVR1 expression in GC tissues, ACVR1 could be a promising target for GC immunotherapy [49].

Fc gamma receptor IIIa (FCGR3A)-type and CD14-type macrophages, as well as pertinent prognostic variables, were identified using single-cell data to help predict the prognosis and response to immunotherapy in patients with STAD. A single-cell atlas for STAD patients was generated in this preliminary investigation, and the nomogram of macrophages and the associated gene signature showed promising immunotherapy and prognostic prediction capabilities [50].

Immune checkpoint inhibitors work well, but patient selection requires more accurate indicators. Another potential predictive biomarker for the effectiveness of immunotherapy is LINC00862. The expression of LINC00862 exhibited a strong predictive value for the efficacy of immunotherapy and was correlated with immune cell infiltration and immune checkpoint expression. The cause of LINC00862 overexpression in gastric and cervical cancer is superenhancer activity. LINC00862 may be a prognostic and diagnostic biomarker [51]. Age, stage, and risk score were found to be independent predictive markers for patients with GC according to multivariate Cox analysis. The immune checkpoint inhibitor response can be predicted by examining the immunophenotype and tumor mutation burden of individuals in the low-risk group. Individuals with lower risk scores for both PD-1+ and cytotoxic T lymphocyte antigen4+ showed greater sensitivity to immunotherapy [52]. The prediction signature of the pathomics-driven ensemble model continued to be an independent predictor for progression-free survival in patients with GC who underwent immunotherapy when multivariable Cox regression was adjusted for clinicopathological variables, such as the sex, age, carcinoembryonic antigen, carbohydrate antigen 19-9, therapy regime, line of therapy, differentiation, location, and PD-L1 expression in all patients (*p* < 0.001, hazard ratio (HR) 0.35 (95% CI 0.24 to 0.50)). Pathogenomic studies revealed that the immune system, cancer, and metabolism-related molecular pathways drive the ensemble model. It also shows correlations with immune-related features such as immune score, tumor purity, and the estimation of stromal and immune cells in malignant tumor tissues using the expression data score. The reaction to ICIs utilizing H and E-stained whole-slide images (WSIs) was predicted with great accuracy and resilience by a pathomics-driven ensemble model. As a result, it might be a useful and innovative tool for precision immunotherapy [53]. The Gastric Cancer Immune Prognostic Score (GCIPS) is for patients with GC receiving immune checkpoint inhibitor (ICI) treatment who have a poor prognosis. Survival analyses revealed a significant association (all with *p* < 0.05) between the GCIPS and overall survival (OS) and progression-free survival (PFS). Moreover, an independent predictive factor for both PFS and OS was shown to be the GCIPS. The analyses of the validation set provided additional evidence of the stability and dependability of the GCIPS for patient prognosis prediction. In conclusion, nomograms that integrate the GCIPS demonstrated excellent precision in both the validation and test sets. Furthermore, the nomograms showed that the TNM stage was not the only component with a lower predictive value than the GCIPS [54].

## 5. Other Factors

### 5.1. Imaging Factors

Biomarkers discovered using CT imaging are connected to the invasion of M1 macrophages, which are traditionally active macrophages. Because CT imaging biomarkers are linked to innate immune signaling, M1 macrophage infiltration, and the ability to accurately predict the results of immunotherapy in GC patients, they may be used to inform specific treatment decisions [55].

By integrating radiomics and deep learning analysis, a noninvasive method for predicting the TME status from radiological images was created. The approach predicts the clinical response in patients receiving checkpoint blockade immunotherapy, and, when paired with current biomarkers, it further increases prediction accuracy [56].

### 5.2. TME Factors

The structure of the microbial microenvironment is an essential feature influencing the development of tumors. Based on bacterial lipopolysaccharide (LPS)-related hub genes, an LPS-related hub gene (LRHG) signature was established that can help predict immunotherapy efficacy and patient prognosis in patients with GC [57]. In GC, there is a strong correlation between the TME and the expression of histone deacetylases (HDACs). Glutathione peroxidase 4 (GPX4) expression in tumor cells and inhibition of the macrophage migration inhibitory factor (MIF) signaling pathway in the TME may be crucial tactics for synergistic cold tumor immunotherapy for GC [58]. One common microenvironmental component in solid tumors is hypoxia. The hypoxia score is an important tool for determining the prognosis of GC patients and directing pharmacological treatments since it has a negative correlation with immune cell infiltration [59].

### 5.3. Clinical Factors

The following common risk variables were observed to significantly affect patient survival at one-, three-, and five-year intervals in GC patients after radical surgery and immunotherapy: advanced age; carbohydrate antigen 72-4 (CA72-4) level; carbohydrate antigen 125 (CA125); carcinoembryonic antigen (CEA) level; tumor lymph node metastases; tumor invasion; multiple tumors; tumor peripheral nerve invasion (PNI); tumor size; and *H. pylori* infection [60]. Other clinical factors were identified late. Serum iron can be used as an indicator to predict the reaction to immune checkpoint inhibitors (ICIs) and is favorably correlated with their efficacy in advanced metastatic malignancies by promoting innate immunity and cytokine production [61]. When GC patients are receiving treatment with an ICI, the development of sarcopenia or myosteatosis is a good indicator of how well they will fare clinically [62]. Furthermore, patient prognosis can be predicted using high and low values of the GC-Integrated Oxidative Stress Score (GIOSS), a biomarker reflecting systemic oxidative stress in the body [63].

### 5.4. Extracellular Factors

One essential molecular chaperone for protein folding, intracellular distribution, and controlling tumor biological behavior in the extracellular environment is heat shock protein 90 (HSP90). HSP90 levels in GC patients who had received fluorouracil/platinum-based advanced first-line treatment were inversely correlated with short-term effectiveness. Patients with GC had significantly higher plasma HSP90 levels after first-line treatment failed [64].

Low blood levels of several tumor suppressor microRNAs (miRNAs) have been linked to poor outcomes and tumor growth in a number of cancer types, according to recent studies. The results of test-scale and large-scale analyses showed that patients with GC had considerably lower plasma levels of miR-5193 than healthy volunteers (HVs). Reduced blood concentrations of miR-5193 are linked to the worsening of GC and its course, and patients with GC may benefit from nucleic acid immunotherapy [65].

The connection between GC cells and the TME is promoted by exosomes, which are essential transporters. The innate targeting abilities, durability, and biocompatibility of these nanoscale extracellular vesicles make them attractive therapeutic candidates for the treatment of GC. Nevertheless, there are still a lot of issues to be resolved before exosomes are widely used as medication delivery vehicles [66].

## 6. Conclusions

Numerous studies on immunotherapy for GC have been conducted to identify predictive factors for the benefits of immunotherapy. All predictive factors of immunotherapy efficacy in GC patients—programmed cell death factors, genetic factors, immunological factors, immune factors, imaging factors, TME factors, and clinical factors—are the subject of research. We advise that these studies be given great attention since there is currently no viable cure for GC.

## Figures and Tables

**Table 1 diagnostics-14-01247-t001:** Major developments in programmed cell death predictive factors research.

Programmed Cell Death Predictive Factors of Immunotherapy in Gastric Cancer	References
- disulfidptosis-related gene prognostic risk model	[4]
- M2 macrophages were strongly correlated with a high PANscore	[5]
- predictive signature AC129926.1, AC023511.1, AP002954.1, LINC01537, and TMEM7 based on CRLs	[6]
- ARLSig has strong predictive value for the survival of GC patients	[7]
- PRS was created using the TCGA, GSE15459, GSE26253, GSE62254, and GSE84437 datasets	[8]
- PI3K pathway linked to PD-L1 positivity is linked to increased immunotherapy efficacy	[9]
- potentially valid serum predictive markers for identifying individuals who might profit from PD-1 inhibitors paired with chemotherapy include IL-6, IL-2, IL-17A, and NLR	[10]
- PD-L1+CD68+ macrophages may be a viable prognostic indicator in primary GC patients	[11]
- ITGB1 may be an effective prognostic biomarker and a useful predictor of outcome for GC patients receiving anti-PD-1 medication	[12]
- a reduced tumor mass, absence or small number of lymph node metastases, and a large combined positive score; neoadjuvant chemotherapy plus PD-1 antibodies for the prediction of pCR in AGC patients undergoing neoadjuvant chemotherapy combined with PD-1 antibody immunotherapy	[13]
- memory PD-1+CD8+ T cells and the ratio of PD-1+CD8+ T cells to PD-1+CD4+ T cells may be useful for identifying individuals who would benefit from immunotherapy among AGC patients	[14]
- the CD4+/CD8+ ratio may be a potential predictive indicator for patients under PD-1 inhibitor-based combination therapy for advanced gastric and esophageal cancer, and it can independently predict dermatological damage	[15]
- MFSD2A may function as a prognostic biomarker for the response to anti-PD-1 immunotherapy in AGC patients	[16]
- the peripheral CD4+ T-cell subset has shown significant predictive value for therapeutic response and longer survival in AGC patients	[17]
- patients with GC receiving combination immunotherapy are predicted to respond better when HSPA4 is upregulated	[18]
- the combination of Tregs, Ki-67, and age (65 years or older) can more accurately predict the likelihood of unfavorable responses	[19]
- the incidence of irAEs could serve as a stand-in marker for ICIs	[20]
- utilizing biomarkers based on the numbers of various lymphocyte subpopulations, it is possible to predict the clinical prognosis and efficacy of immunotherapy in patients with AGC	[21]
- a negative correlation between CRABP2 and the immune checkpoint markers PD-1, PD-L1, and CTLA-4 was observed	[22]

**Table 2 diagnostics-14-01247-t002:** Major developments in genetic predictive factors research.

Genetic Predictive Factors of Immunotherapy in Gastric Cancer	References
- age and the GARGs were found to be separate risk factors for GC	[23]
- the E2F score is a useful predictor of survival and responsiveness to treatment	[24]
- the low-risk group of GC patients (with genes involved in the DNA repair pathway) are likely to benefit more from immunotherapy	[25]
- suppressing DNAAF3 reduces GCs’ capacity for migration and proliferation as well as their ability to activate T lymphocytes inside the TME	[26]
- STK40 is the most important gene linked to senescence that influences the development of GC	[27]
- CS might represent the therapeutic impact of immunotherapy	[28]
- MFAP2+ CAFs create an immune-evading TME with immune effector cells that are debilitated	[29]
- the immunotherapy characteristics of the TME were strongly correlated with two genes (MSH4 and RPL22L1) and four methylation probes (EPM2AIP1|MLH1:cg27331401, LNP1:cg05428436, and TSC22D2:cg15048832)	[30]
- a lncRNA signature model associated with DNA methylation regulators is a novel method for predicting the treatment response and stratification of patients with GC	[31]
- N6-methyladenosine (m6A) regulator-mediated genes in the glycolytic pathway predict the prognosis and immunotherapy response of GC patients	[32]
- the adenosine pathway-based signature is a potentially useful risk stratification tool that might inform personalized prognostication and immunotherapy for GC patients	[33]
- the TIDE score and the IPS indicate that decreased immunotherapy benefit is associated with elevated expression of CHSY3	[34]
- FIRGs have significant therapeutic and predictive potential for GC and have a role in the onset and progression of the disease	[35]
- TGF-beta, TP53, and NRF2 pathways dominated the high-risk group, whereas the LRTK-RAS and WNT pathways represented the low-risk group	[36]
- the Anoikis-related lncRNA genes predictive model has been shown to have high predictive accuracy and may be used as a guide for prognostic evaluation and clinical treatment of patients with STAD	[37]
- five methylation-related genes, namely, CHAF1A, CPNE8, PHLDA3, SPARC, and EHF were found to potentially be significant for patient prognosis	[38]
- Hedgehog signaling, a highly conserved system that controls cell division and growth, is crucial in the development of STAD	[39]
- SPP1 has been identified as a stand-alone predictor of STAD prognosis and may control the disease course	[40]
- the T-cell-related genes CD5, ABCA8, SERPINE2, ESM1, SERPINA5, and NMU have been suggested as prospective targets for immunotherapy in patients with STAD and can aid in prognosis estimation	[41]
- ICB treatment for EBVaGC may be better guided by the identification of SMARCA4, TMB, and CTLA-4 mutations as possible predictive markers of ICB success	[42]

**Table 3 diagnostics-14-01247-t003:** Major developments in immunological predictive factors research.

Immunological Predictive Factors of Immunotherapy in Gastric Cancer	References
- the BMRG signature has proven accurate and reliable in the prediction of prognosis in GC patients	[43]
- CLEC11A may be a predictive and immunological biomarker	[44]
- the INHBA is a potentially useful indicator of immunotherapy response	[45]
- the CRG signature could be a reliable and effective tool for prognostic prediction and therapeutic guidance	[46]
- NK cell marker genes point to possible treatment targets	[47]
- CSK may predict immunotherapy and molecular targeting in addition to offering suggestions for a treatment plan	[48]
- ACVR1 could be a promising target for GC immunotherapy	[49]
- FCGR3A-type and CD14-type macrophages were identified using single-cell data to help predict the prognosis and response to immunotherapy in patients with STAD	[50]
- LINC00862 may be a prognostic and diagnostic biomarker	[51]
- patients with lower risk scores for both PD-1+ and cytotoxic T lymphocyte antigen4+ showed greater sensitivity to immunotherapy	[52]
- the prediction signature of the pathomics-driven ensemble model continued to be an independent predictor for progression-free survival in patients with GC who underwent immunotherapy	[53]
- GCIPS is an independent predictive factor for both PFS and OS	[54]

## Data Availability

Not applicable.

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
