# Peer review of "Predictive Factors of Immunotherapy in Gastric Cancer: A 2024 Update"

_diagnostics, 2024, doi:10.3390/diagnostics14121247_

Round 1
Reviewer 1 Report
Comments and Suggestions for Authors
The Review describes predictive factors for GC immunotherapy. It is an interesting and well-structured Review, however, there are several minor issues:
- Authors should add more information in the Abstract
- In the “Other factors” authors should add any extracellular factors affecting the response to immunotherapy ( for example, extracellular heat shock proteins or mRNAs or exosomes, etc)
- Line 63- “the expression of positive programmed cell death.. “. Authors should remove the word “positive”
- Line 57 - «Certain immune-activated cells and immune-activated cells had ..” is confusing
- Line 298 and lines 301 and 302. Authors should proofread this paragraph
- Overall the manuscript requires extensive proofreading
Author Response
1.Authors should add more information in the Abstract
We redid the abstract!
2.In the “Other factors” authors should add any extracellular factors affecting the response to immunotherapy ( for example, extracellular heat shock proteins or mRNAs or exosomes, etc)
We talked about extracellular heat shock proteins, mRNAs and exosomes at the end of the article:
“5.4. Extracellular factors
One essential molecular chaperone for protein folding, intracellular distribution, and controlling tumor biological behavior in the extracellular environment is heat shock protein 90 (HSP90). HSP90 levels in GC patients who had received fluorouracil/platinum-based advanced first-line treatment were inversely correlated with short-term effectiveness. Patients with GC had significantly higher plasma HSP90 levels after first-line treatment failed (64).
Low blood levels of several tumor suppressor microRNAs (miRNAs) have been linked to poor outcomes and tumor growth in a number of cancer types, according to recent studies. The results of test-scale and large-scale analyses showed that patients with GC had considerably lower plasma levels of miR-5193 than healthy volunteers (HVs). Reduced blood concentrations of miR-5193 are linked to the worsening of GC and its course, and patients with GC may benefit from nucleic acid immunotherapy (65).
The connection between GC cells and the TME is promoted by exosomes, which are essential transporters. The innate targeting abilities, durability, and biocompatibility of these nanoscale extracellular vesicles make them attractive therapeutic candidates for the treatment of GC. Nevertheless, there are still a lot of issues to be resolved before exosomes are widely used as medication delivery vehicles (66).”
3.Line 63- “the expression of positive programmed cell death.. “. Authors should remove the word “positive”
We removed the word positive’
4.Line 57 - «Certain immune-activated cells and immune-activated cells had ..” is confusing
We have redone the sentence:
“Programmed cell death (PCD)-related signature (PRS) was created using TCGA, GSE15459, GSE26253, GSE62254, and GSE84437 datasets through an integrative process that used ten machine learning techniques.”
5.Line 298 and lines 301 and 302. Authors should proofread this paragraph
We redid the paragraph:
“Immune checkpoint inhibitors work well, but patient selection requires more accurate indicators. Another potential predictive biomarker for the effectiveness of immunotherapy is LINC00862. The expression of LINC00862 exhibited a strong predictive value for the efficacy of immunotherapy and was correlated with immune cell infiltration and immune checkpoint expression. The cause of LINC00862 overexpression in gastric and cervical cancer is superenhancer activity. LINC00862 may be a prognostic and diagnostic biomarker”
6.Overall the manuscript requires extensive proofreading
We have re-read the article and corrected the highlighted wrong aspects.
Reviewer 2 Report
Comments and Suggestions for Authors
This is an interesting article about the use of immunotherapy in gastric cancers. The authors did a research of the specialized literature and revealed the results of the studies that were published during 2023 and 2024. So all the informations are new and relevant.
My suggestion is to put together some data in tables, in order to make everything easier to read (for each category- genetic factors, immunological factors etc.).
Author Response
Thank you for the appreciations. We created 3 tables in which we highlighted the major research and discoveries.